# JAK Inhibitors in Cutaneous T-Cell Lymphoma: Friend or Foe? A Systematic Review of the Published Literature

**DOI:** 10.3390/cancers16050861

**Published:** 2024-02-21

**Authors:** Seyed Mohammad Vahabi, Saeed Bahramian, Farzad Esmaeili, Bardia Danaei, Yasamin Kalantari, Patrick Fazeli, Sara Sadeghi, Nima Hajizadeh, Chalid Assaf, Ifa Etesami

**Affiliations:** 1School of Medicine, Tehran University of Medical Sciences, Tehran 1461884513, Iran; mohammadvahabi73@gmail.com (S.M.V.); y-kalantari@student.tums.ac.ir (Y.K.); 2School of Medicine, Isfahan University of Medical Sciences, Isfahan 8174673461, Iran; saeedbahramian97@gmail.com; 3School of Medicine, Shahid Beheshti University of Medical Sciences, Tehran 1985717411, Iran; farzad.esmaeili.t@sbmu.ac.ir (F.E.); bdanaei75@gmail.com (B.D.); 4Independent Researcher, Agoura Hills, CA 91301, USA; patrickfazeli@gmail.com; 5School of Medicine, Iran University of Medical Sciences, Tehran 1449614535, Iran; sarasadeghimd@gmail.com (S.S.); hajizadeh.n@tak.iums.ac.ir (N.H.); 6Department of Dermatology and Venerology, Helios Klinikum Krefeld, 47805 Krefeld, Germany; 7Institute for Molecular Medicine, Medical School Hamburg, 20457 Hamburg, Germany; 8Departments of Dermatology, Razi Hospital, School of Medicine, Tehran University of Medical Sciences, Tehran 1983969411, Iran

**Keywords:** cutaneous T-cell lymphomas, CTCL, Janus kinase, JAK inhibitor, mycosis fungoides, subcutaneous panniculitis-like T-cell lymphoma, SPTCL, JAK/STAT pathway

## Abstract

**Simple Summary:**

Cutaneous T-cell lymphomas (CTCLs) are a rare kind of skin cancer that currently has no curative treatment except allogeneic stem cell transplantation. The aim of this systematic review is to investigate the effectiveness of Janus kinase inhibitor drugs as a treatment for CTCLs. This study showed that Janus kinase inhibitors can be effective in the treatment of CTCLswith acceptable adverse effects. Adverse events have been reported especially in patients with immunosuppression or an underlying autoimmune disease. We recommend conducting more studies, especially clinical trials, to investigate the benefits of these drugs for the treatment of cutaneous T-cell lymphomas.

**Abstract:**

Cutaneous T-cell lymphomas (CTCLs) are a group of lymphoid neoplasms with high relapse rates and no curative treatment other than allogeneic stem cell transplantation (allo-SCT). CTCL is significantly influenced by disruption of JAK/STAT signaling. Therefore, Janus kinase (JAK) inhibitors may be promising for CTCL treatment. This study is a systematic review aiming to investigate the role of JAK inhibitors in the treatment of CTCL, including their efficacy and safety. Out of 438 initially searched articles, we present 13 eligible ones. The overall response rate (ORR) in the treatment with JAK inhibitors in clinical trials was 11–35%, although different subtypes of CTCL showed different ORRs. Mycosis fungoides showed an ORR of 14–45%, while subcutaneous-panniculitis-like T-cell lymphoma (SPTCL) displayed an ORR ranging from 75% to 100%. Five cases were reported having a relapse/incident of CTCL after using JAK inhibitors; of these, three cases were de novo CTCLs in patients under treatment with a JAK inhibitor due to refractory arthritis, and two cases were relapsed disease after graft-versus-host disease treatment following allo-SCT. In conclusion, using JAK inhibitors for CTCL treatment seems promising with acceptable side effects, especially in patients with SPTCL. Some biomarkers, like pS6, showed an association with better responses. Caution should be taken when treating patients with an underlying autoimmune disease and prior immunosuppression.

## 1. Introduction

Cutaneous T-cell lymphomas (CTCLs) are a group of lymphoid neoplasms which firstly affect the skin [1]. CTCLs are usually common in older adults but rarely seen in children, with an incident rate of 7.5 per one million people annually [2,3]. Mycosis fungoides (MF), Sézary syndrome (SS), primary cutaneous CD30+ lymphoproliferative disorders (LPDs), and their subtypes account for more than 85% of CTCLs. Although there are further subtypes of CTCL, like subcutaneous panniculitis-like T-cell lymphoma (SPTCL) and primary cutaneous γ/δ T-cell lymphoma, these subtypes are rare (Table 1) [1,4,5]. CTCL diagnosis is formed on clinical manifestations and correlation to histopathology. The prognosis of MF and SS could be predicted by the cutaneous lymphoma international prognostic index (CLIPi), although it needs further modifications [6,7]. MF progression risk varies from 12% in the early stages up to nearly 80% in the late stages [8]. There are several therapies available, including topical corticosteroids, phototherapy, antibodies such as brentuximab vedotin or mogamulizumab, interferons, radiotherapy, and chemotherapy, and their use is based on the stage of the disease [9,10,11,12]. However, till now there are no curative treatments available other than allogeneic stem cell transplantation (allo-SCT), which has been shown to produce long-term complete remission in some groups of patients [13].

The JAK/STAT signaling pathway, central to the pathophysiology of CTCL, is frequently dysregulated through JAK1, JAK3, STAT3, and STAT5B point mutations and copy number gains in JAK2, STAT3, and STAT5B [14,15,16,17]. These genetic aberrations lead to overactive signaling, driving uncontrolled cell proliferation and survival, characteristic of the disease [18,19]. Of note, the specific JAK/STAT pathway proteins that might be affected by different mutations vary in different subtypes of CTCL [20,21,22], as shown in Figure 1. Janus kinase (JAK) inhibitors offer a targeted therapeutic approach, aiming to suppress this dysregulation by preventing the phosphorylation and activation of JAK and STAT proteins, thereby impeding the progression of CTCL [23,24].

As a biological treatment, JAK inhibitors are a class of drugs that act on four proteins: JAK1, JAK2, JAK3, and TYK2. They are able to control the inflammatory process because they activate intracytoplasmic transcription factors like signal transducer and activator of transcription (STAT). As a result, upon activation, they enter the nucleus and form dimers, which either positively or negatively regulate genes [25,26]. These drugs have recently been approved for inflammatory diseases such as atopic dermatitis, graf-versus-host disease (GVHD), hidradenitis suppurativa, and alopecia areata [17,27,28].

The use of JAK inhibitors is not limited to dermatologic diseases and is also FDA-approved for other diseases such as ulcerative colitis, myelofibrosis, and rheumatoid arthritis [29,30,31]. Several JAK inhibitors have been approved for different types of diseases. JAK inhibitors could be divided into groups by their mechanism of action. For example, types I and II bind to the ATP-binding site of the JAKs, but allosteric JAK inhibitors bind to a site other than the ATP-binding site in JAKs. Also, these drugs target different JAKs: upadacitinib, filgotinib, and oclacitinib target JAK1; baricitinib, abrocitinib, and ruxolitinib target JAK1 and 2; tofacitinib targets JAK1, 2, and 3; and peficitinib is a pan-JAK inhibitor [30].

As different studies described different features of the diseases like different JAK/STAT mutations in CTCL patients [32,33,34], the results of some in vitro and clinical studies showed that CTCL patients may benefit from treatment with JAK inhibitors or show the reversal of side effects [22,35,36,37].

Therefore, the aim of this review is to inspect the potential role of JAK inhibitors in cutaneous T-cell lymphoma treatment.

## 2. Materials and Methods

This study is a systematic review of clinical trials, case series, and case reports, aiming to investigate the role of JAK inhibitors in the treatment of CTCL, their efficacy, and their possible side effects. The main aim of our systematic review was to evaluate the efficacy and safety of JAK inhibitors in the treatment of CTCL based on the published literature. This review is dedicated to documenting the diagnosis, patient number, dosage, treatment duration, outcomes, and side effects of JAK inhibitors in various CTCL settings. It adheres to the “Preferred Reporting Items for Systematic Reviews and Meta-Analyses” (PRISMA) statement [38]. The protocol has not been registered.

The search strategy involved querying the Web of science, Medline, Embase, Scopus, ClinicalTrials.gov, and WHO ICTRP databases using keywords related to JAK inhibitors, cutaneous T-cell lymphoma, and mycosis fungoides. The search was conducted on 3 December 2023 and included all records up to that date. A manual search was also performed in gray literature (Figure 2).

To exclude unrelated papers, five reviewers (PF, SB, BD, NH, and SS) screened the papers by title, abstract, and full text, independently. Another reviewer (IE), made the final decision in case of disagreement. The inclusion criteria encompassed studies on patients with CTCL, studies on patients treated with at least one JAK inhibitor, and studies conducted solely on humans. This review does not include studies on patients with other types of cutaneous lymphoma such as B-cell lymphoma, studies on animals, in vivo studies, or reports of using JAK inhibitors with no effect. Three studies [39,40,41] were part of another study [35,37]; so, we will not present them separately.

Using the National Heart, Lung, and Blood Institute (NHLBI) quality assessment tools for clinical trials, the quality of the included clinical trials was assessed by three reviewers. This tool is a 14-question checklist that assesses the quality of articles in terms of methodology, report of the findings, and possible distortions. Another reviewer, SMV, made the final decision in case of disagreement.

Data extraction involved collecting information on the dosage, duration, outcomes, and side effects of JAK inhibitors in various CTCL settings from the included articles, along with the patients’ number, age group and demographic information on the studies, such as data collection period, country, and study design.

## 3. Result

In this review, we present thirteen articles that showed the clinical effects of JAK inhibitors on CTCLs. Among them, nine papers (Table 2 and Table 3) mentioned the positive effects of JAK inhibitors on CTCL, and four papers (Table 4) mentioned the adverse effects, including relapse or de novo of CTCL following the administration of a JAK inhibitor. The studies comprised two non-randomized phase II clinical trials, eight case reports, and three case series.

### 3.1. Studies Evaluating the Efficacy of JAK Inhibitors in the Treatment of CTCL

#### 3.1.1. Clinical Trials

Horwitz et al. [35] evaluated cerdulatinib (a JAK1, 2, and 3 inhibitor), 30 mg, twice-daily efficacy in a phase II trial with 98 patients with refractory/relapsed CTCL or peripheral T cell lymphoma (PTCL), who at least received one prior systemic therapy. The patients were treated until progression, intolerance, or adequate response to allo-SCT. All patients receive antimicrobial prophylaxis. The median age was 65 (21–85) years for PTCL patients and 62 (24–80) years for CTCL patients. The overall response rate (ORR) of 60 patients in the PTCL group evaluated after treatment was 35% (21 patients). The ORR of the CTCL patients was 35% (9), with the highest recorded for the MF patients (45%; 9% of the patients had a complete response (CR)), and the lowest for the SS patients (17%, without CR). The CTCL patients showed rapid pruritus improvement, without an association with tumor response. For both groups, the median response duration was pending, but some patients showed a persistent response lasting more than 12 months. The most common treatment-emergent grade 3+ adverse events were an increase in lipase and amylase levels (21.5% and 17.3%, respectively) without clinical pancreatitis, neutropenia (8.1%), diarrhea (8.1%), anemia (7.1%), and fatigue (6.1%). Grade 3+ infections happened in 28 patients (29%) without complications.

Moskowitz et al. [37] conducted a study on ruxolitinib (a JAK1 and 2 inhibitor), 20 mg, twice-daily efficacy in patients with refractory/relapsed PTCL or MF. Fifty-three patients were divided into three biomarker-defined groups (group 1, with JAK1-, JAK2-, JAK3-, STAT3-, or STAT5B-activating mutations; group 2, with functional evidence of JAK/STAT activation but lack of mutations in the above-mentioned factors; group 3, who did not meet the criteria for group 1 or 2) and received treatment until disease progression. The primary endpoint was the clinical benefit rate (CBR), defined as the combination of stable disease (SD) lasting at least 6 months, partial response (PR), and CR. For the MF and subcutaneous panniculitis-like T-cell lymphoma (SPTCL) patients, the CBR was 14% and 100%, respectively. The MF patients reported a CBR of 14% (ongoing PR > 18 months in one of seven patients). The PTCL patients in groups 1, 2, and 3 showed CBRs of 53% (10 patients), 45% (5 patients), and 13% (2 patients), respectively (*p* = 0.73). Eight patients had a CBR > 12 months. The expression of phosphorylated S6, a marker of phosphoinositide 3-kinase (PI3K) or mitogen-activated protein kinase activation, in <25% of the tumor cells was associated with a response to ruxolitinib (*p* = 0.05). Treatment-related serious adverse events in different patients included HSV-1 stomatitis (1.9%), SBP (1.9%), febrile neutropenia (5.7%), anemia (1.9%), and herpes zoster (1.9%).

#### 3.1.2. Case Reports/Series

Levy et al. [42] presented a 16-year-old boy with a relapse of hemophagocytic lymphohistiocytosis (HLH) and histologically confirmed SPTCL. The boy had mild to diffuse panniculitis with subcutaneous enhancement, lymphopenia, persistence of HLH features, episodic recurrent fever, and pain. After receiving different treatments with no improvement, a 15 mg twice-daily treatment with ruxolitinib (a JAK1 and 2 inhibitor) was initiated, led to rapid improvement, and made it possible to stop other medications after two months, without relapse. Four months after ruxolitinib initiation, the patient was in remission. Then, he decided to stop ruxolitinib and, eight months later, he was admitted to the hospital with relapsing SPTCL. The administration of a 20 mg twice-daily ruxolitinib monotherapy rapidly cleared the fever and caused the alleviation of HLH features and panniculitis. Ten months later, under ruxolitinib treatment, he was completely in remission of HLH and SPTCL.

Castillo et al. [43] reported a man in his 80s with generalized (80% of his body surface) scaling and erythema and a palpable lymph node in his right inguinal area. Initial skin biopsies were inconclusive, and different therapies were used with no significant response. With an initial diagnosis of atopic dermatitis, upadacitinib (a JAK1 inhibitor) treatment at 15 mg daily was initiated. Scales, erythema, pruritus, and axillary and pelvic indurated plaques rapidly improved, but he still had lymphadenopathy in his right inguinal area, with no change. Two months later, with another set of biopsies, the patient was finally diagnosed as having MF stage 3 (T4, N0, M0, B0), and the initial AD was considered a misdiagnosis. Sixteen weeks after starting the upadacitinib treatment, he showed a CR, with less than 10% of his skin involved.

Kook et al. [44] presented a 43-year-old man with erythematous patches on his trunk and a lichenified pinkish plaque with excoriation on his lower back. He had had these lesions for seven years and had been treated with different medications, following the diagnosis of atopic dermatitis. A second biopsy showed an aggregation of atypical lymphocytes, and the diagnosis was corrected to MF. After 16 weeks of receiving upadacitinib (a JAK1 inhibitor), the skin lesion on the trunk disappeared, leaving a faint hyperpigmentation. 

Watson et al. [45] presented a 19-year-old man with a left thigh erythematous painful edema, fever, pancytopenia, splenomegaly, hyperferritinemia, elevated serum CD25, hypertriglyceridemia, hypofibrinogenemia, and hemophagocytosis in a biopsy of his bone marrow. A skin biopsy confirmed SPTCL. After receiving different therapies, he first showed remission but then relapsed. Without a suitable donor for allo-SCT, prednisolone and 15 mg twice-daily ruxolitinib (a JAK1 and 2 inhibitor) were initiated. He achieved improvement after one month, and the corticosteroids were tapered, without a relapse of the disease. Seven weeks later, he presented with severe neutropenia, presumed to be ruxolitinib-related. Ruxolitinib was stopped, and prednisolone was started. Within two days, he showed an acute increase in serum ferritin and extensive subcutaneous activity but remained asymptomatic. He underwent allo-SCT. He remained in complete remission for 15 months post allo-SCT.

Hansen et al. [46] presented a 33-year-old man with a lower extremity nodular rash, fever, and weight loss. With a diagnosis of HLH and biopsy-proven SPTCL, he received several drugs (methotrexate, prednisone, doxorubicin, cyclophosphamide, vincristine, dexamethasone, and etoposide) without improvement. Then, he received etoposide, intravenous immunoglobulin, and ruxolitinib (a JAK1 and 2 inhibitor). Because of coagulopathy and retroperitoneal hemorrhage, ruxolitinib was stopped after 5 days, and he received alemtuzumab without improvement. Then, he received ruxolitinib and low-dose etoposide which led to HLH remission within four weeks. After four weeks, etoposide was stopped, and ruxolitinib was continued for 10 months at 15 mg twice daily and then at 10 mg daily and was finally discontinued. Also, glucocorticoids were tapered within seven months since ruxolitinib initiation. Although different infections complicated his status, the patient was well after one year.

Duan et al. [47] analyzed, in a retrospective study, 18 children with SPTCL (pathologically proven) to compare the clinical efficacy of a multi-drug chemotherapy–immunomodulatory therapy. Of these patients, four, with a mean age of 9.5 years, received ruxolitinib (a JAK1 and 2 inhibitor) in combination with immunomodulatory agents. Three patients (75%) showed CR, with a mean follow-up time of 11 months. Four months after ruxolitinib withdrawal, one patient (25%) had a recurrent mass and underwent autologous hematopoietic stem cell transplantation.

Zhang et al. [48] reported the efficacy of ruxolitinib (a JAK1 and 2 inhibitor) against HLH and panniculitis manifestations in six children. The median age at onset was 10.5 years (0.8–12.4). All patients developed HLH, and five presented different lesions of the skin, such as ulcerations, plaques, and subcutaneous nodules. Skin biopsies showed that three of them had panniculitis in the absence of evidence of lymphoma, but two of them had SPTCL. With unfavorable responses to several medications, ruxolitinib (a JAK1 and 2 inhibitor) was initiated and showed rapid disease resolution with/or remission maintenance in the long term. Of those with SPTCL, a twelve-year-old boy showed a growing subcutaneous nodule and a high fever. After achieving partial remission with ruxolitinib 10 mg twice daily, he was discharged. One month later, he returned with a relapse. Ruxolitinib (10 mg twice daily) and methylprednisolone were started, and he showed improvement. After two weeks, methylprednisolone was stopped, and ruxolitinib was stopped after 15 months. Later, for two weeks, he initiated ruxolitinib (5 mg daily) himself because of constitutional symptoms, which resulted in disease resolution. In his last visit, he had no relapse seven months after stopping the treatment. The second patient, a 5-year-old boy, had a waist mass with a recurrent fever and subcutaneous nodules. The administration of ruxolitinib 5 mg twice daily led to a rapid CR for HLH, without relief of the subcutaneous nodules. Adding methylprednisolone for three weeks caused the elimination of the skin nodules. With low-dose ruxolitinib monotherapy and no HLH activation, he complained of the reappearance of new subcutaneous masses. No improvement was seen with twice-daily 2.5 mg ruxolitinib. Then, methylprednisolone was added, which made the skin masses disappear. After two months, he underwent allo-SCT, and 13 months after allo-SCT, he was in CR for SPTCL and HLH.

### 3.2. De Novo CTCL following JAK Inhibitor Treatment 

Iinuma et al. [49] presented a 74-year-old woman with a previous history of rheumatoid arthritis (RA) who had been treated with several drugs including sulfasalazine, methotrexate, iguratimod, prednisolone, and tocilizumab with only moderate response. Then, with peficitinib (a JAK1, JAK2, JAK3, and TYK2 inhibitor), 150 mg daily for one year, she showed only a partial response. Therefore, the treatment was switched to upadacitinib (a JAK1 inhibitor), 7.5 mg daily. Two weeks later, she developed multiple red-to-brown crusted papules on her extremities, without systemic symptoms or lymphadenopathy. A skin biopsy established the diagnosis of lymphomatoid papulosis (LyP). Upadacitinib was stopped, and the skin lesions were treated with topical corticosteroids. Within a month, the skin lesions resolved, with hyperpigmentation and without any reported relapse. 

Knapp et al. [50] presented a 42-year-old man with previous inflammatory arthritis and erythema elevatum diutinum (EED). After showing no improvement following the administration of several drugs such as sulfasalazine, mycophenolate mofetil, adalimumab, and dapsone, he received tofacitinib (a JAK1, 2, and 3 inhibitor). Eight weeks later, he presented with a right periorbital edema and new skin lesions, and two weeks later, with involvement of the inguinal folds. Tofacitinib was stopped, but 2 weeks later, he had a severe right periorbital edema and several violaceous firm papules on the inguinal folds, the extremities, and the trunk. Biopsies of the skin confirmed the diagnosis of LyP. Based on the new diagnosis of LyP, he received methotrexate (15 mg weekly), which at a 12-week follow-up showed cutaneous and ocular findings’ clearance.

Saito et al. [51] presented a 78-year-old man who received different drugs, including adalimumab, bucillamine, betamethasone, methotrexate, and tocilizumab, for seronegative rheumatoid arthritis. Then, baricitinib (a JAK1 and 2 inhibitor) was initiated, while continuing betamethasone. Seven months later, he developed severe itch and erythroderma on his extremities and trunk, without peripheral lymphadenopathy and an elevated leucocyte count and lymphocyte count. Based on laboratory tests, histopathology of the skin, and immunohistochemistry and tomography findings, an SS diagnosis was made. Baricitinib was discontinued, and mogamulizumab was initiated, 1 mg weekly for eight times within 11 weeks. This was followed by treatment with narrowband ultraviolet B phototherapy in combination with bexarotene, and a PR was observed after 3 months.

Cohen et al. [52] presented two cases of CTCL worsening after receiving ruxolitinib (a JAK1 and 2 inhibitor) for chronic GVHD treatment. One was a man in his 60s, with T3N0M0B0 MF and large-cell transformation’ histological evidence. He received multiple treatments, including PUVA therapy, topical clobetasol, acitretine, bexarotene, brentuximab-nivolumab, romidepsin, etoposide–ifosfamide, and bendamustine, and underwent allo-SCT, which resulted in CR. With a muscular GVHD diagnosis after reporting muscle weakness and myalgias, he received intravenous immunoglobulin, corticosteroids, and ruxolitinib. His skin condition worsened, and he developed voluminous skin tumors with erythroderma. A histological analysis showed CTCL relapse, and ruxolitinib was tapered. He showed a PR after receiving total skin electron-beam therapy and chemotherapy. The other case was a woman with folliculotropic MF in her 50s and large-cell transformation (T3N0M0B0) who underwent allo-SCT after receiving multiple treatments (chlormethine, carmustine, PUVA therapy, and bexarotene), which resulted in a CR. After some months, corticosteroids and ruxolitinib were started for her severe sclerotic and ulcerated skin due to chronic GVHD. Then, she showed a transformed MF cutaneous relapse, which was histopathologically proved. In addition, a flow cytometry analysis showed blood involvement, leading to the diagnosis of SS. Ruxolitinib was ceased, and treatment with brentuximab vedotin and liposomal doxorubicin was initiated, leading to a blood and clinical PR.

## 4. Discussion

In the past decades, treatments based on the immune system, such as targeted therapy and antibody therapy, have made significant progress. Mogamulizumab, an anti-CCR4 antibody, and brentuximab vedotin, a conjugated antibody against CD30, are FDA-approved for cutaneous T-cell lymphomas and showed good efficacy in previous studies [53,54].

Further studies with novel agents are ongoing, e.g., targeting the CD47–SIPRa axis, i.e., the “don´t eat me” signal. It was shown that CD47 is frequently overexpressed in CTCL cells and that the interaction with anti-CD47 antibodies induces phagocytosis of the malignant cells by macrophages but also modulates cells of the tumor microenvironment. Moreover, a recent clinical trial using intralesionally TTI-621, a novel CD47 inhibitor, showed promising results, with high activity in patients with relapsed or refractory MF and SS [55,56,57,58].

Important biological processes like immune response, carcinogenesis, cell differentiation, and cell death are mediated by the JAK–STAT signaling pathway by affecting different growth factors and cytokines [16,59,60]. Consequently, medications that disrupt distinct JAK–STAT signaling pathways are used for different diseases, especially autoimmune and auto-inflammatory ones [61]. Eleven JAK and STAT enzymes (STAT1, STAT2, STAT3, STAT4, STAT5A, STAT5B, and STAT6, JAK1, JAK2, JAK3, tyrosine kinase 2) make up the major parts of the JAK–STAT pathway, involving different cell types and transmembrane receptors [16,62].

Recent data showed that the lack of expression of JAK–STAT pathway components plays a role in CTCL pathogenesis and prognosis. Since STAT4 and STAT6 expression is inversely regulated in CTCL, the loss of STAT4 can be both a potent diagnostic tool for leukemic CTCL/SS and a poor prognostic marker for early MF [23]. A higher expression of JAK3 is associated with the severity of CTLC malignancy [34]. STAT3 activation is recognized as a risk factor for MF progression [63].

Considering that the JAK–STAT signaling cascade plays a major role in CTCL pathogenesis and may therefore be a promising target for CTCL treatment, the evaluation of different biomarkers and genes and their targeting would be useful for improving CTCL treatment options [22,36,37].

There is a limited number of prospective clinical trials using JAK inhibitors in CTCL patients, with MF and SS patients commonly studied. In general, the response rate was different for the patients due to the disease subtype and was higher for patients receiving cerdulatinib (JAK1, 2, and 3 inhibitor) instead of ruxolitinib (JAK1, 2 inhibitor), with ORRs of 35% and 11%, respectively [35,37]. Although different biomarker groups, such as STAT3-, STAT5B-, JAK1-, JAK2-, or JAK3-activating mutations or the lack of mutation in these factors with JAK–STAT activation functional evidence, did not show an association with the response rate, the expression of pS6 showed an association with a better response to ruxolitinib [37]. S6 is a ribosomal protein that plays a role in PI3K and mammalian target of rapamycin (mTOR) pathways. PI3K and mTOR pathway activation happens in many cancers; so, targeting S6 is a potential strategy for cancer treatment [64]. This implies that targeted therapy would be beneficial in CTCL treatment.

Also, studies with JAK inhibitors showed similar results in PTCL patients. A clinical trial showed golidocitinib (a JAK-1 inhibitor) efficacy and safety in PTCL patients as a maintenance therapy. In a group of 30 patients with prior CR and a median follow-up of 8.3 months and in another group of 18 patients with prior PR and a median follow-up of 5.6 months, 82.4% and 66.7% of the patients responded to a maintenance therapy with golidocitinib, respectively [65].

Case reports and case series also demonstrated high effectivity in patients with rare subtypes of CTCL, e.g., SPTCL [42,43,44,45,46,47,48]. Of the ten SPTCL patients described in these studies, nine showed CR after using JAK inhibitors, and one relapsed after stopping JAK inhibitors and underwent allo-SCT [42,45,46,47,48]. This different response rate can be due to differences in the types of mutations and pathways in the IL–JAK–STAT axis compared to other types of CTCL [66].

However, besides studies describing positive effects after the use of JAK inhibitors in CTCL patients, there are also several reports on the de novo development of CTCL [49,50,51,52]. To our knowledge, a total of five cases have been reported, of which, three cases took multiple medications to treat arthritis [49,50,51], and two cases received GVHD treatment following allo-SCT [52]. Since all patients were receiving immunosuppressive medications, it seems that immunosuppression for a long time before taking JAK inhibitors may play a role in increasing the risk of CTCL incidents or relapse.

In our study, except for the reported cases of recurrence or de novo malignancy, the main complications of JAK inhibitor use were neutropenia and infection, which could be managed. Studies on JAK inhibitor use against inflammatory bowel diseases and rheumatological diseases showed that the risk of major adverse cardiovascular events (MACEs), cancer, and infection increased in these patients [67,68,69]. However, a systematic review and meta-analysis by Ingrassia et al. [70] did not show an increased risk of MACEs in patients using JAK inhibitors for dermatologic diseases. It seems that different factors, like the underlying disease and the duration of the treatment, play a role in terms of MACE risk [67]. Caution should be taken, especially when treating older patients with a history of cardiac diseases.

## 5. Conclusions

Using JAK inhibitors for CTCL seems promising, with acceptable side effects, indicating that it is worth investigating them in larger prospective randomized clinical trials. Especially, SPTCL seems to be the ideal entity for treatment using JAK inhibitors, showing a high response rate with CR. Some biomarkers, like pS6, showed an association with better responses. However, caution should be taken with patients with an underlying autoimmune disease and prior immunosuppression. Further studies, especially larger-scale clinical trials, are needed to investigate the efficacy and safety of JAK inhibitors for CTCL treatment.

## Figures and Tables

**Figure 1 cancers-16-00861-f001:**
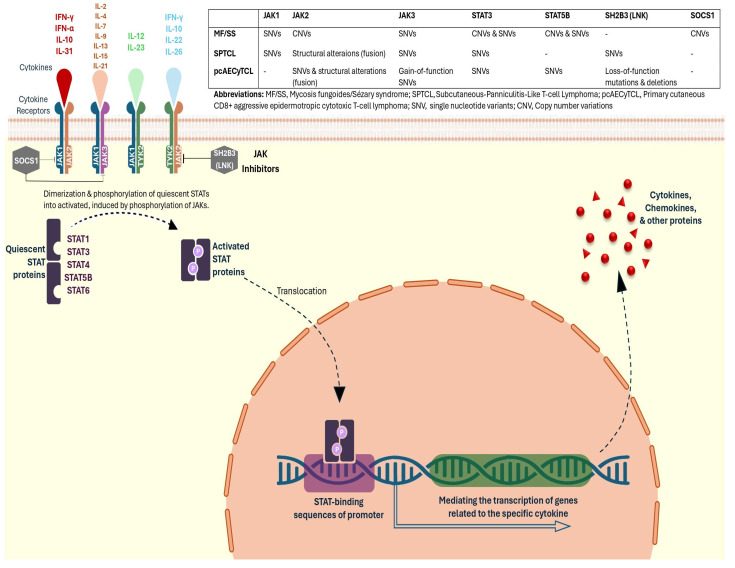
Disruptions in the JAK/STAT pathway are linked to various subtypes of CTCL, each mostly impacted by distinct genetic mutations. The pathway becomes active when cytokines engage their respective cell surface receptors, triggering the activation of JAKs. In CTCL, the mutation spectrum varies across subtypes: for MF and SS, JAK1 and JAK3 exhibit single-nucleotide variants (SNVs), while STAT3 and STAT5B are similarly affected by SNVs. In SPTCL, JAK2 is particularly notable for SNVs. Primary cutaneous CD8+ aggressive epidermotropic cytotoxic T-cell lymphoma (pcAECyTCL) does not show JAK mutations but is characterized by SNVs in JAK2, JAK3, STAT3, and STAT5B. Additionally, increased expression of these STAT proteins is frequently correlated with copy number gains, particularly in JAK2 for MF/SS and in both STAT3 and STAT5B across the subtypes mentioned. Copy number variations (CNVs) and structural alterations such as fusions in these genes can lead to enhanced or constitutive activation of the pathway, driving the progression of CTCL. Moreover, the negative regulatory roles of SOCS1 and SH2B3 (LNK) are highlighted, where CNVs or loss-of-function mutations can lead to unregulated JAK/STAT pathway activity, emphasizing their potential as therapeutic targets. For instance, loss-of-function mutations and deletions in SOCS1 could contribute to the aberrant activation of the JAK/STAT pathway in CTCL subtypes. The detailed mutation profile for each subtype underscores the complexity and the necessity for subtype-specific therapeutic approaches, including the use of JAK inhibitors to correct the dysregulated signaling in these lymphomas. “P” stands for phosphate; triangles and squares stand for cytokines, chemokines and other proteins. Abbreviations: IFN-γ, interferon gamma; IL, interleukin; SOCS1, suppressor of cytokine signaling-1; LNK, lymphocyte adapter protein; JAK, Janus kinase; TYK2, tyrosine kinase-2; STAT, signal transducer and activator of transcription.

**Figure 2 cancers-16-00861-f002:**
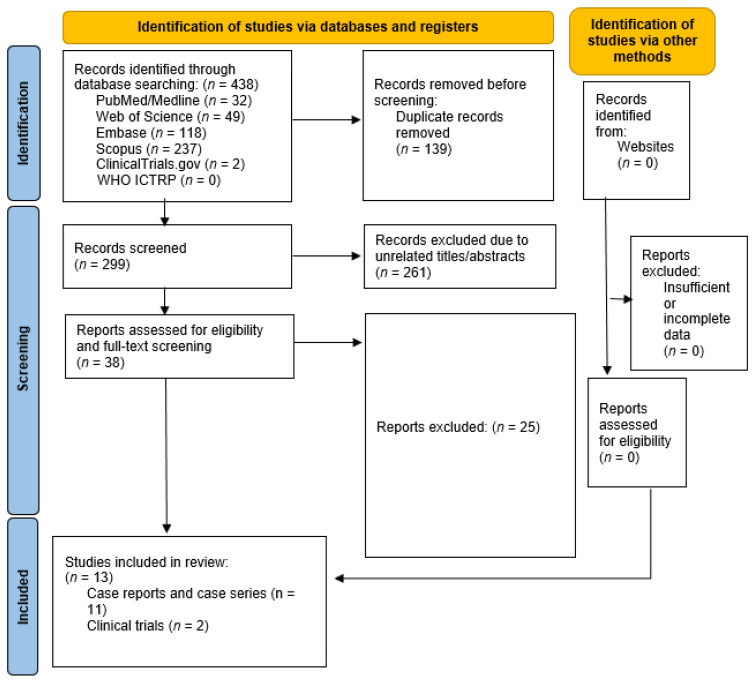
Preferred Reporting Items for Systematic Reviews and Meta-Analyses (PRISMA). Flow chart of the number of studies identified and selected into the systematic review and meta-analysis.

**Table 1 cancers-16-00861-t001:** Primary CTCL subtypes’ characteristics.

CTCL Subtype	Frequency (%)	5-Years Disease-Specific Survival (%)	Clinical Features	T-Cell Phenotype
Mycosis fungoides (MF) Folliculotropic MF Pagetoid reticulosis Granulomatous slack skin	39 5 <1 <1	88 75 100 100	Patches and plaques; (ulcerating) tumors in advanced stage	CD3+, CD4+, CD8−
Sézary syndrome (SS)	2	36	Triad of pruritic erythroderma, generalized lymphadenopathy, and clonally related neoplastic T cells with cerebriform nuclei (Sézary cells) in the skin, lymph nodes, and peripheral blood	CD4+, CD7−, CD26−
Primary cutaneous CD30+ lymphoproliferative disorders (LPDs) Primary cutaneous anaplastic large lymphoma (C-ALCL) Lymphomatoid papulosis (LyP)	8 12	95 99	Solitary or localized nodules or tumors Chronic course of recurrent, self-healing papulonecrotic, or nodular skin lesions.	CD3+/−, CD4+, CD8−, CD30+ CD4+, CD8− or CD4−, CD8+ or CD4−, CD8−
Subcutaneous panniculitis-like T-cell lymphoma	1	87	Subcutaneous nodules and plaques	CD3+, CD4−, CD8+
Primary cutaneous peripheral T cell lymphoma, rare subtypes Primary cutaneous gamma-delta T cell lymphoma (PCGD-TCL) Primary cutaneous aggressive epidermotropic CD8+ T cell lymphoma (PCAECyTCL) Primary cutaneous CD4+ small- or medium-sized LPDs Primary cutaneous acral CD8+ LPD	<1 <1 6 <1	11 31 100 100	Ulcerating plaques and tumors Ulcerating plaques, nodules, and tumors Solitary nodule or tumor on the face or upper trunk Solitary papule or nodule on acral site (ear; nose)	CD3+, CD4−, CD8−/+ CD3+, CD4−, CD8+ CD3+, CD4+, CD8−, CD279/PD-1+ CD3+, CD4−, CD8+
Primary cutaneous peripheral T cell lymphoma, not otherwise specified	2	15	Localized skin lesions	CD4+

**Table 2 cancers-16-00861-t002:** Studies evaluating the efficacy of JAK inhibitors in the treatment of CTCL; clinical trials.

Study	Design	Patient Number	Diagnosis	Drug Name	Dosage	Duration	Outcome	Side Effect
Horwitz et al., USA 2019 [35]	Clinical trial	37	CTCL	Cerdulatinib JAK1, 2, and 3 inhibitor	30 mg twice daily	NA	ORR was 35% (13/37) MF (ORR of 45%; 9% of the patients achieved CR) versus SS (ORR of 17%, with no CR).	Present
Moskowitz et al., USA 2021 [37]	Clinical trial	10	7 MF 1 SPTCL 1 PCALCL 1 PCGDTCL	Ruxolitinib JAK1 and 2 inhibitor	20 mg twice daily	NA	ORRs for MF, SPTCL, and PCALCL were 14% (1/7), 100%, and 100%, respectively.	Present

CTCL: cutaneous T-cell lymphoma, MF: mycosis fungoides, SS: Sézary syndrome, SPTCL: subcutaneous panniculitis-like T-cell lymphoma, PCGDTCL: primary cutaneous γ/δ T-cell lymphoma, PCALCL: primary cutaneous anaplastic large cell lymphoma. ORR: overall response rate, CR: complete response, NA: Not accessible, Present: Showed side effect which discussed in text.

**Table 3 cancers-16-00861-t003:** Studies evaluating the efficacy of JAK inhibitors in the treatment of CTCL; case reports and case series.

Study	Design	Patient Number	Diagnosis	Drug Name	Dosage	Duration	Outcome	Side Effect
Levy et al., France 2020 [42]	Case report	1	SPTCL + HLH	Ruxolitinib JAK1 and 2 inhibitor	15 mg twice daily + 20 mg twice daily	4 + 14 months	CR	NA
Castillo et al., USA 2022 [43]	Case report	1	Erythrodermic MF	Upadacitinib JAK1 inhibitor	15 mg daily	16 weeks	CR	NA
Kook et al., South Korea 2022 [44]	Case report	1	MF	Upadacitinib JAK1 inhibitor	NA	16 weeks	CR	NA
Watson et al., Australia 2022 [45]	Case report	1	SPTCL	Ruxolitinib JAK1 and 2 inhibitor	15 mg twice daily	7 weeks	CR	Present
Hansen et al., Canada 2020 [46]	Case series	1	SPTCL+ HLH	Ruxolitinib JAK1 and 2 inhibitor	15 mg twice daily then 10 mg twice daily	11 months	CR	Present
Duan et al., China 2022 [47]	Case series	4	SPTCL	Ruxolitinib JAK1 and 2 inhibitor	NA	NA	75% (3/4) CR 25% (1/4) allo-SCT	NA
Zhang et al., China 2023 [48]	Case series	2	SPTCL + HLH	Ruxolitinib JAK1 and 2 inhibitor	10 mg twice daily 5 mg twice daily	16 months About 8 months	CR CR	NA

SPTCL: subcutaneous panniculitis-Like T-cell lymphoma, HLH: hemophagocytic lymphohistiocytosis, MF: mycosis fungoides. CR: complete response, allo-SCT: allogeneic stem cell transplantation. NA: Not accessible, Present: Showed side effect which discussed in text.

**Table 4 cancers-16-00861-t004:** Studies reporting relapses or incidents of CTCL following JAK inhibitor treatment.

Study	Design	Patient Number	Diagnosis	Drug Name	Dosage	Duration	Outcome	Side Effect
Iinuma et al., Japan 2022 [49]	Case report	1	Rheumatoid arthritis	Upadacitinib JAK1 inhibitor	7.5 mg daily	2 weeks	Incident of LyP	NA
Knapp et al., USA 2022 [50]	Case report	1	Erythema elevatum diutinum and associated inflammatory arthritis	Tofacitinib JAK1, 2, and 3 inhibitor	NA	10 weeks	Incident of LyP	NA
Saito et al., Japan 2023 [51]	Case report	1	Seronegative rheumatoid arthritis	Baricitinib JAK1 and 2 inhibitor	NA	7 months	Incident of SS	NA
Cohen et al., France 2023 [52]	Case report	2	Chronic graft-versus-host disease	Ruxolitinib JAK1 and 2 inhibitor	20 mg daily 30 mg daily	NA NA	CTCL relapse MF relapse	NA

LyP: lymphomatoid papulosis, SS: Sézary syndrome, CTCL: cutaneous T-cell lymphoma, MF: mycosis fungoides. NA: Not accessible.

## Data Availability

Data sharing is not applicable to this article, as no new data were created or analyzed in this study.

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
