# Peer review of "JAK Inhibitors in Cutaneous T-Cell Lymphoma: Friend or Foe? A Systematic Review of the Published Literature"

_cancers, 2024, doi:10.3390/cancers16050861_

Round 1

Reviewer 1 Report

Comments and Suggestions for Authors

The systematic review "JAK inhibitors in cutaneous T-cell lymphoma: Friend or Foe? A 2 systematic review of the published literature" is an important and well compiled work highlighting potential for JAK inhibitors as therapeutic avenue in cutaneous -T cell lymphoma. Authors used very well defined methodological description for database and trial selection, inclusion and exclusion criteria, although the included clinical trials have small n, whether any RCT is available or not is not clear. Few minor methodological addition if possible by the authors can improve the quality of the manuscript further.

1. As it is systematic review in methodology section a hypothesis or a priory could be included.

2. if possible authors can provide a table of key words used for database search and a Boolean chart for search terms.

3, authors used NHLBI quality assessment tool for CTs , did authors use considered using Jaded score or TC Chalmers scoring for literature? Authors can provide list of 14 criterions used by NHLBI.

4.Will it be plausible to prepare a forest plot based on odds ratio, this is not mandatory but does provide strong statistical clue regarding efficacy  of JAK inhibitors in cutaneous -T cell lymphoma.

Author Response

The Responses (R) to Comments (C) of Reviewer

Reviewer #1

C: Authors used very well-defined methodological description for database and trial selection, inclusion and exclusion criteria, although the included clinical trials have small n, whether any RCT is available or not is not clear. Few minor methodological addition if possible by the authors can improve the quality of the manuscript further.

R: Thanks for your insightful comment. Unfortunately, until now, no RCT regarding Efficacy and safety of JAK inhibitors for CTLCL treatment has been published. There are two clinical trials included in our study. Both are non-randomized but prospective phase II-trials. Based on your valuable comment, we clarified this information in the result and discussion part.

C1: As it is systematic review in methodology section a hypothesis or a priory could be included.

R: Thank you for your comment. A hypothesis was added to the methodology section line 126-127 and has been highlighted. “The main hypothesis of our systematic review is to evaluate the efficacy and safety of JAK inhibitors in the treatment of CTCL based on the published literature.”

C2: If it possible authors can provide a table of key words used for database search and a Boolean chart for search terms.

R: Thanks for your valuable comment. We included a table of keywords used in our search strategy as a supplementary file. Also Figure 2 is a chart that shows our search based on databases. Instead, this review was conducted in accordance with PRISMA and the Cochrane Handbook for Systematic Reviews (Higgins JPT, Green S, Cochrane Collaboration. Cochrane Handbook for Systematic Reviews of Interventions. Chichester: Wiley-Blackwell; 2008).

C3: Authors used NHLBI quality assessment tool for CTs, did authors use considered using Jaded score or TC Chalmers scoring for literature? Authors can provide list of 14 criterions used by NHLBI.

R: We provided a list of 14 criterions used by NHLBI and cited it in Methodology section. (https://www.nhlbi.nih.gov/health-topics/study-quality-assessment-tools).

C4: Will it be plausible to prepare a forest plot based on the odds ratio? This is not mandatory but does provide strong statistical clue regarding efficacy of JAK inhibitors in cutaneous T-cell lymphoma.

R: Thank you for your insightful comment. Due to the diverse methodologies of the included articles and the limited sample size, conducting a meta-analysis and creating a forest plot is in this case not feasible.

Reviewer 2 Report

Comments and Suggestions for Authors

This is a systematic review about JAK inhibitors and cutaneous lymphomas. It is well written, it is easy to read and to understand. I am not sure if the type of selected manuscripts allow to make a meta-analysis with forest plot. Possibly not as there is not enough scientific evidence. To improve the manuscript, the authors may consider the following comments:

(1) Lines 46-50. Regarding the classification of cutaneous T-cell lymphomas. Only mycosis fungoides (+Sezary) and CD30+LPD are highlighted. But there are other types of cutaneous TCLs. Could you please add a table in this paragraph or in appendix with the names of the subtypes and a brief description (including diagnosis, treatment, and prognosis?

For example, relevant subtypes are the following:

Cutaneous T cell and NK cell lymphomas
Mycosis fungoides
Mycosis fungoides variants and subtypes
    Folliculotropic mycosis fungoides
    Pagetoid reticulosis
    Granulomatous slack skin
Sézary syndrome
Adult T cell leukemia/lymphoma
Primary cutaneous CD30+ lymphoproliferative disorders
    Primary cutaneous anaplastic large cell lymphoma
    Lymphomatoid papulosis
Subcutaneous panniculitis-like T cell lymphoma
Extranodal NK/T cell lymphoma, nasal type
Primary cutaneous peripheral T cell lymphoma, rare subtypes
    Primary cutaneous gamma-delta T cell lymphoma
    Primary cutaneous aggressive epidermotropic CD8+ T cell lymphoma (provisional)
    Primary cutaneous CD4+ small/medium-sized pleomorphic T cell lymphoproliferative disorder (provisional)
    Primary cutaneous acral CD8+ T cell lymphoma (provisional)
Primary cutaneous peripheral T cell lymphoma, not otherwise specified

***WHO-EORTC classification for primary cutaneous lymphomas. Blood 2019; 133:1703.

(2) Have you revised the recent updates in lymphoma classification? Please refer and cite these references, if necessary.

The International Consensus Classification of Mature Lymphoid Neoplasms: a report from the Clinical Advisory Committee. https://doi.org/10.1182/blood.2022015851

The 5th edition of the World Health Organization Classification of Haematolymphoid Tumours: Lymphoid Neoplasms. doi: 10.1038/s41375-022-01620-2. Epub 2022 Jun 22.

(3) Line 53. You may want to highlight that In the Western world, cutaneous T cell lymphomas (CTCL) represent approximately 75 percent of all primary cutaneous lymphomas. Mycosis fungoides and primary cutaneous CD30+ lymphoproliferative disorders account for approximately 90 percent of all CTCL. Other types of CTCL are rare and may run a very aggressive clinical course.

**In line 49 only Myc fungoides and CD30+ are mentioned, but prognosis of Mycosis fungoides is not to aggressive.

(4) Is it worth mentioning as well Cutaneous Lymphoma International Prognostic Index (CLIPi), and the CLIC Prognostic Index?

(5) Could you please modify figure 1? Some letters are too small and cannot be read without increasing the size of the pdf file in the screen.

(6) In Figure 1 there is a table. Could you please add the abbreviations for MS/SS, SPTCL, pcAECyTCL, SNVs, CNVs, etc.?

(7) In the introduction. Could you please described briefly the different types of JAK inhibitors?Tofacitinib, baricitinib, upadacitinib, filgotinib, peficitinib, ruxolitinib, abrocitinib, ritlecitinib, others (deucravacitinib), etc. Of note, some of them affect only some specific JAKs such as abrocitinib that is JAK1 inhibitor.

(8) Additionally, please also mention that  are being used for RA, other subtypes of arthritis, ulcerative colitis, etc.?

(9) Secondary effect of JAK inhibitors could also be described briefly.

(10) Have you included the systematic review into prospero database?

(11) Do the type of data obtained from this review allow to calculate and draw a forest plot?

https://www.nature.com/articles/s41433-021-01867-6

(12) Line 325. Do CTLs express CD47? Expressed by neoplastic T cells, or the microenvironment?

(13) Line 374. Regarding "may be used as a monotherapy or combination therapy". Is the use as monotherapy approved by the FDA?

(14) Please comment on the increased risk of major adverse cardiovascular events (MACE).

(15) Please refer to this publication

Nature Reviews Rheumatology. Winthrop KL. The emerging safety profile of JAK inhibitors in rheumatic disease. Nat Rev Rheumatol 2017; 13:234. Copyright © 2017. https://www.nature.com/nrrheum/.

Author Response

The Responses (R) to Comments (C) of Reviewer

Reviewer #2

C1: Lines 46-50. Regarding the classification of cutaneous T-cell lymphomas. Only mycosis fungoides (+Sezary) and CD30+LPD are highlighted. But there are other types of cutaneous TCLs. Could you please add a table in this paragraph or in appendix with the names of the subtypes and a brief description (including diagnosis, treatment, and prognosis?

For example, relevant subtypes are the following:

Cutaneous T cell and NK cell lymphomas
Mycosis fungoides
Mycosis fungoides variants and subtypes
    Folliculotropic mycosis fungoides
    Pagetoid reticulosis
    Granulomatous slack skin
Sézary syndrome
Adult T cell leukemia/lymphoma
Primary cutaneous CD30+ lymphoproliferative disorders
    Primary cutaneous anaplastic large cell lymphoma
    Lymphomatoid papulosis
Subcutaneous panniculitis-like T cell lymphoma
Extranodal NK/T cell lymphoma, nasal type
Primary cutaneous peripheral T cell lymphoma, rare subtypes
    Primary cutaneous gamma-delta T cell lymphoma
    Primary cutaneous aggressive epidermotropic CD8+ T cell lymphoma (provisional)
    Primary cutaneous CD4+ small/medium-sized pleomorphic T cell lymphoproliferative disorder (provisional)
    Primary cutaneous acral CD8+ T cell lymphoma (provisional)
Primary cutaneous peripheral T cell lymphoma, not otherwise specified

***WHO-EORTC classification for primary cutaneous lymphomas. Blood 2019; 133:1703.

R: Thank you for your valuable comment. According to your esteemed opinion, we added other subtypes of primary CTCL in the new Table 1, mentioning subtypes of primary CTCL with a brief description.

C2: Have you revised the recent updates in lymphoma classification? Please refer and cite these references, if necessary. The International Consensus Classification of Mature Lymphoid Neoplasms: a report from the Clinical Advisory Committee. https://doi.org/10.1182/blood.2022015851

The 5th edition of the World Health Organization Classification of Haematolymphoid Tumours: Lymphoid Neoplasms. doi: 10.1038/s41375-022-01620-2. Epub 2022 Jun 22.

R: Thank you for your comment. We revised it as shown in table 1. and cited these articles.

C3: Line 53. You may want to highlight that In the Western world, cutaneous T cell lymphomas (CTCL) represent approximately 75 percent of all primary cutaneous lymphomas. Mycosis fungoides and primary cutaneous CD30+ lymphoproliferative disorders account for approximately 90 percent of all CTCL. Other types of CTCL are rare and may run a very aggressive clinical course. In line 49 only Mycosis fungoides and CD30+ are mentioned, but prognosis of Mycosis fungoides is not too aggressive.

R: Thanks for your valuable comment. We made this information now in the revised version clear that MF is in most cases indolent compared to some rare subtypes.

Line 58-59: MF progression risk varies from 12% in the early stages up to nearly 80% in the late stages.

C4: Is it worth mentioning as well Cutaneous Lymphoma International Prognostic Index (CLIPi), and the CLIC Prognostic Index?

R: Thanks for your comment. It was added and highlighted in the introduction part.

Line 57-58: Prognosis of MF and SS could be predicted by the cutaneous lymphoma international prognostic index (CLIPi) although it needs further modifications.

C5: Could you please modify figure 1? Some letters are too small and cannot be read without increasing the size of the pdf file in the screen.

R: Agreed. Thank you for your diligent comment. As per your kind request, the Figure.1 has been modified in a way that the fonts are now enlarged.

C6: In Figure 1 there is a table. Could you please add the abbreviations for MS/SS, SPTCL, pcAECyTCL, SNVs, CNVs, etc.?

R: Thank you for your careful comment. As per your kind request, the abbreviations for MS/SS, SPTCL, pcAECyTCL, etc. have been added at the footnote of the table section of the figure, within the Figure.1 itself. Additionally, the abbreviations for all other terms, demonstrated in the graphic section of the figure have all been expanded at the end of the figure legend (Line98-100).

C7: In the introduction. Could you please described briefly the different types of JAK inhibitors?Tofacitinib, baricitinib, upadacitinib, filgotinib, peficitinib, ruxolitinib, abrocitinib, ritlecitinib, others (deucravacitinib), etc. Of note, some of them affect only some specific JAKs such as abrocitinib that is JAK1 inhibitor.

R: Thank you. This is explained and highlighted now in a new paragraph in the introduction section.

Line 111-116: JAK inhibitors could be divided into groups by mechanism of action. For example, Types I and II bind to the ATP-binding site of the JAKs but allosteric JAK inhibitors bind to a site other than the ATP-binding site in the JAKs. Also, these drugs target different JAKs: Upadacitinib, Filgotinib, and Oclacitinib target JAK 1; Baricitinib, Abrocitinib, Ruxolitinib targets JAK 1 and 2; Tofacitinib targets JAK 1, 2, and 3; and Peficitinib is a Pan-JAK inhibitor.

C8: Additionally, please also mention that are being used for RA, other subtypes of arthritis, ulcerative colitis, etc.?

R: Thank you. This is explained and highlighted now in a new paragraph in the introduction section.

Line 108-110: The use of JAK inhibitors is not limited to dermatologic diseases and is also FDA-approved for other diseases such as ulcerative colitis, myelofibrosis, and rheumatoid arthritis. Several JAK inhibitors have been approved for different types of diseases.

C9: Secondary effect of JAK inhibitors could also be described briefly.

R: Thank you for mentioning this. A new paragraph was added and highlighted in the discussion part about JAK inhibitors side effects.

Line 407-416: In our study, except for the reported cases of recurrence or De novo malignancy, the main complications of JAK inhibitor use were neutropenia and infection, which could be managed. Studies on the JAK inhibitors use in inflammatory bowel diseases and rheumatological diseases have shown that the risk of major adverse cardiovascular events (MACE), cancer and infection increase in these patients. However, a Systematic review and Meta-Analysis by Ingrassia et al. did not show an increased risk of MACE in patients using JAK inhibitors for dermatologic diseases. It seems different factors, like the underlying disease and duration of treatment, play a role in terms of MACE risk. Caution should be taken, especially in patients with older ages and a history of cardiac diseases.

C10: Have you included the systematic review into prospero database?

R: Thank you for your valuable comment. Since registering the methodology of systematic reviews in the Prospero database is optional, and would be more beneficial if done at the commencement of our study, including the methodology of the present article in this database is not plausible. As Figure 2 shows, our study follows PRISMA guidelines for systematic review and meta-analysis. (Higgins JPT, Green S, Cochrane Collaboration. Cochrane Handbook for Systematic Reviews of Interventions. Chichester: Wiley-Blackwell; 2008).

C11: Do the type of data obtained from this review allow to calculate and draw a forest plot?

R: Thanks for your comment. Because of the heterogeneity of included articles, only two prospective clinical trials and the rest are only case reports, in several aspects like their methodologies and also the limited number of included articles in this study, it is not possible for us to conduct a meta-analysis and illustrate the data as a forest plot.

C12: Line 325. Do CTLs express CD47? Expressed by neoplastic T cells, or the microenvironment?

R: Thank you for your insightful comment. CD47 is a glycoprotein expressed on many normal cells that inhibits phagocytosis by macrophages via interaction with SIRP-alpha on the macrophage. The occurrence of its expression in CTCL and its therapeutical role based on current trails and perspective for the future is now mentioned in the revised version We extended and highlighted this comment.

Line 352-358: Further studies with novel agents are ongoing e.g. targeting CD47-SIPRa axis the “don´t eat me signal”. It has been shown that CD47 is frequently overexpressed in CTCL-cells and that the interaction with anti-CD47 antibodies induces phagocytosis of the malignant cells by macrophages but also modulates cells of the tumor microenvironment. Moreover, a recent clinical trial using intralesionally TTI-621, a novel CD47 inhibitor, showed promising results with high activity in patients with relapsed or refractory MF and SS.

C13: Line 374. Regarding "may be used as a monotherapy or combination therapy". Is the use as monotherapy approved by the FDA?

R: Thanks for your insightful comment. JAK inhibitors are neither approved by FDA-nor by the EMEA for CTCL. This point was only discussed based on previously published papers that were included in this study. We edited it to be clear.

 Line 418-420: Using JAK inhibitors for CTCL seems promising with acceptable side effects, indicating that it is worth to investigate them in larger prospective randomized clinical trials.

C14: Please comment on the increased risk of major adverse cardiovascular events (MACE).

R: Thank you for your precise comment. This part was added to the discussion section and has been highlighted.

Line 407-416: Studies on the JAK inhibitors use in inflammatory bowel diseases and rheumatological diseases have shown that the risk of major adverse cardiovascular events (MACE), cancer and infection increase in these patients. However, a Systematic Review and Meta-Analysis by Ingrassia et al. did not show an increased risk of MACE in patients using JAK inhibitors for dermatologic diseases. It seems different factors, like the underlying disease and duration of treatment, play a role in terms of MACE risk. Caution should be taken, especially in patients with older ages and a history of cardiac diseases.

C15: Please refer to this publication Nature Reviews Rheumatology. Winthrop KL. The emerging safety profile of JAK inhibitors in rheumatic disease. Nat Rev Rheumatol 2017; 13:234. Copyright © 2017. https://www.nature.com/nrrheum/.

R: This article was used and cited in a new paragraph in the introduction section.

Line 108-110: The use of JAK inhibitors is not limited to dermatologic diseases and is also FDA-approved for other diseases such as ulcerative colitis, myelofibrosis, and rheumatoid arthritis.